# Differences in SARS-CoV-2 antibody prevalence at the end of the pre-vaccination period between age groups: A cross-sectional analysis of the multi-ethnic population-based HELIUS study

Renee Bolijn[1], Annemieke M. W. Spijkerman[2], Henrike Galenkamp[1]*, Anneke Blokstra[2], Liza Coyer[3,4], Anders Boyd[3,4,5], Maria Prins[3,4], Karien Stronks[1]

1 Department of Public and Occupational Health, Amsterdam UMC location University of Amsterdam, Amsterdam, The Netherlands, 2 Centre for Prevention, Lifestyle and Health, National Institute for Public Health and the Environment, Bilthoven, The Netherlands, 3 Department of Infectious Diseases, Public Health Service of Amsterdam, Amsterdam, The Netherlands, 4 Department of Infectious Diseases, Amsterdam UMC location University of Amsterdam, Amsterdam, The Netherlands, 5 Stichting HIV Monitoring, Amsterdam, The Netherlands

* h.galenkamp@amsterdamumc.nl

**Data Availability Statement:** The HELIUS data are owned by the Amsterdam University Medical

## Abstract

### Background

During the first waves of the COVID-19 pandemic, SARS-CoV-2 antibody prevalence (seroprevalence) was lower in older compared to younger adults. We studied age group differences in SARS-CoV-2 seroprevalence, across ethnic groups, and assessed the explanatory value of factors that increase the exposure to the virus, and factors related to susceptibility, given the level of exposure.

### Methods

We analysed cross-sectional data from 2,064 participants from the multi-ethnic HELIUS study (Amsterdam, The Netherlands). SARS-CoV-2 seroprevalence at the second wave of infections was compared between age groups (<40, 40–54, and $\geq$55 years), within ethnic groups, using Poisson regression with robust standard errors. To determine whether age group differences were explained by differences in exposure and in susceptibility, we sequentially adjusted for exposure (job setting, occupation level, health literacy, household size, household member with suspected infection), education level, and susceptibility (vitamin D intake, BMI, systolic blood pressure, haemoglobin level, number of comorbidities).

### Results

SARS-CoV-2 seroprevalence did not statistically differ across age groups (p>0.05), but age patterns varied across ethnic groups. Age group differences in SARS-CoV-2 seroprevalence were most pronounced in the Dutch majority group, with the highest prevalence ratio

Center, Amsterdam, The Netherlands. Any researcher can request the data by submitting a proposal to the HELIUS Executive Board as outlined at http://www.heliusstudy.nl/en/researchers/collaboration, by email: heliuscoordinator@amsterdamumc.nl. The HELIUS Executive Board will check proposals for compatibility with the general objectives, ethical approvals and informed consent forms of the HELIUS study. There are no other restrictions to obtaining the data and all data requests will be processed in the same manner.

**Funding:** The HELIUS study is conducted by the Amsterdam University Medical Center, location AMC, and the Public Health Service of Amsterdam. Both organizations provided core support for HELIUS. The HELIUS study is also funded by the Dutch Heart Foundation, the Netherlands Organization for Health Research and Development (ZonMw), European Union (FP-7), and the European Fund for the Integration of non-EU immigrants (EIF). The current study was funded by the National Institute for Public Health and the Environment (RIVM) as part of the COVID-19 research programme commissioned by the Ministry of Health, Welfare and Sports. The funders had no role in study design, data collection and analysis, decision to publish, or preparation of the manuscript.

**Competing interests:** The authors declare that they have no competing interests.

**Abbreviations:** ANOVA, Analysis of Variance; ATC, Anatomical therapeutic chemical; BMI, Body mass index; BP, Blood pressure; CI, Confidence interval; CKD, Chronic kidney disease; COPD, Chronic obstructive pulmonary disease; CVD, Cardiovascular disease; ELISA, Enzyme-linked immunosorbent assay; Hs-CRP, High sensitive c-reactive protein; IgA, IgM, IgG, Immunoglobulin A, M, G; PR, Prevalence ratio; SBSQ, Set of brief screening questions.

in the youngest group (2.55, 95%CI 0.93–6.97) and the lowest in the oldest group (0.53, 95%CI 0.16–1.74), compared to the middle-aged group. In ethnic minority groups, age group differences were smaller. In all groups, patterns did not substantially change after adjustments for exposure and susceptibility variables.

## Conclusion

We found no evidence of age group differences in SARS-CoV-2 seroprevalence, particularly for ethnic minority groups, even when accounting for exposure and susceptibility. While early prevention strategies particularly aimed at protecting older adults against SARS-CoV-2 infections, seroprevalence was similar across age groups in ethnic minority groups. Thus, older adults in ethnic minority groups may pose a target group for additional prevention strategies for future infectious disease outbreaks.

## Introduction

Since December 2019, a large proportion of the world's population has been infected by the SARS-CoV-2 virus [1]. In the early stages of the pandemic, when SARS-CoV-2 vaccines were still under development, differences in occurrence of SARS-CoV-2 infections between population groups, such as age groups, were substantial. Previous studies reported that SARS-CoV-2 antibody prevalence (seroprevalence), which is the proportion of the population who have antibodies to SARS-CoV-2 infection, is generally lower in older adults compared to younger and middle-aged adults [2–4]. Particularly in the absence of widespread vaccination roll-out, insights into factors related to age group differences in SARS-CoV-2 seroprevalence may help to identify specific population groups vulnerable to infection and, ultimately, to improve targeting of prevention strategies, also for future outbreaks of infectious diseases.

It has been hypothesized that younger and middle-aged adults may have a higher likelihood of a SARS-CoV-2 infection because they are more likely to be exposed to the virus. This may be because of their lifestyle with higher intensity of contacts due to social gatherings, occupation, and household size [5]. At the same time, older adults may not only be at higher risk of severe disease and mortality after a SARS-CoV-2 infection [6–8], but may also be more susceptible to infection, after exposure to the virus [9]. This hypothesis is supported by evidence on higher secondary attack rates among adults aged over 60 years compared to younger adults [10], but is also contradicted by others [11]. A potential underlying mechanism for this higher susceptibility to infection among older adults is the aging of the immune system, making older people more susceptible to new pathogens [12]. In addition, older adults may be more susceptible to infection due to their generally poorer health, as indicated by e.g., a higher prevalence of chronic diseases (such as diabetes) and a higher body mass index (BMI) [13]. Furthermore, a lower education level may explain a higher likelihood of infection, since this may be associated with both poorer health [14] and lower uptake of preventive measures [15], and this should be taken into account [16]. Studies testing the exposure hypothesis while also considering potential susceptibility to infection among older adults in the current SARS-CoV-2 pandemic are limited, particularly in multi-ethnic populations.

Large ethnic differences in occurrence of SARS-CoV-2 infections have been reported across countries. Studies from Europe and the USA observed that ethnic minority populations had a higher likelihood of a SARS-CoV-2 infection than majority populations [2, 3, 17–20]. Data

suggest that minority groups were more often exposed to the virus and more susceptible to infection, after exposure. For instance, larger households, having a contact job (i.e., contact with others within 1.5 meters), poorer health, and lower education level are generally more prevalent in ethnic minority groups compared to majority populations, and associated with infection [21–24]. At the same time, age group differences in likelihood of infection may also vary across ethnic groups, for similar reasons as described above [25]. However, studies on differences in likelihood of infection by both age and ethnicity are limited [26–28]. A study from the USA suggests that Black individuals and individuals in the 'other race/ethnic category', particularly at older age, have a higher likelihood of infection compared to white individuals, based on PCR testing data [26]. In line, we previously observed that SARS-CoV-2 seroprevalence was highest in middle-aged persons of Ghanaian origin living in Amsterdam, the Netherlands, after the first wave of the COVID-19 pandemic [27]. Whether this differential likelihood across age groups in minority groups is explained by differences in exposure, susceptibility, or both remains to be investigated.

The aim of this study was 1) to explore differences in SARS-CoV-2 seroprevalence between younger, middle-aged, and older adults in the pre-vaccination period, stratified by six ethnic groups from the population-based HELIUS study. The pre-vaccination period in The Netherlands was defined as the period that started with the initial lockdown (i.e. mid-March [29])) after the first confirmed cases of SARS-CoV-2 in the country on February 27, 2020. Lockdown measures were gradually lifted from mid-May 2020 onwards, after which the second wave started towards the end of August 2020. The lockdown measures applied after this date until December 2020 were not nearly as restrictive as in the first wave. The Dutch vaccination campaign started on January 6, 2021 and SARS-CoV-2 vaccination became available to individuals working in healthcare or > 75 years old during our study period. As such, most HELIUS participants were not yet eligible for vaccination during this study. Against this background, we 2) explored to what extent differences between age groups were explained by differences in factors that increase the exposure to the virus (henceforth: exposure variables), and 3) whether factors related to susceptibility (henceforth: susceptibility variables) played a role in these associations. **Methods**

## Study design and study sample

The current study is part of the Healthy Life in an Urban Setting (HELIUS) study. The HELIUS study is a multi-ethnic prospective cohort study conducted in Amsterdam, the Netherlands, which focuses on (the causes of) the unequal burden of cardiovascular disease, mental health problems and infectious diseases across ethnic groups. Detailed procedures have been previously described [30, 31]. Briefly, HELIUS includes participants of Dutch, Surinamese, Ghanaian, Moroccan, and Turkish origin, aged 18–70 years at inclusion. Potential participants were sampled with a simple random sampling method from the municipality registry, after stratification by ethnicity. Between January 2011 and December 2015, a total of 24,789 individuals were included. Between April 2019 and December 2022, the first follow-up visit took place. During both visits, data were obtained by questionnaire/interview and physical examinations were performed (including the collection of biological samples).

A participant was defined as belonging to one of the included ethnic minority groups if he/she fulfilled one of two criteria: (1) he/she was born outside the Netherlands and has at least one parent born outside the Netherlands (first generation) or (2) he/she was born in the Netherlands but both parents were born outside the Netherlands (second generation) [32]. For the Dutch sample, we invited people who were born in the Netherlands and whose parents were born in the Netherlands. After data collection, Surinamese participants were further classified according to self-reported ethnic origin into "African" and "South-Asian".

The flow of participation selection is shown is Fig 1. In 2020, a SARS-CoV-2 serological substudy consisting of three visits was set up in a subgroup of HELIUS participants who were still in follow-up [17, 27]. In brief, of the 16,845 eligible participants (an eligibility criterion was, amongst others, having provided consent for additional studies), 11,077 were randomly

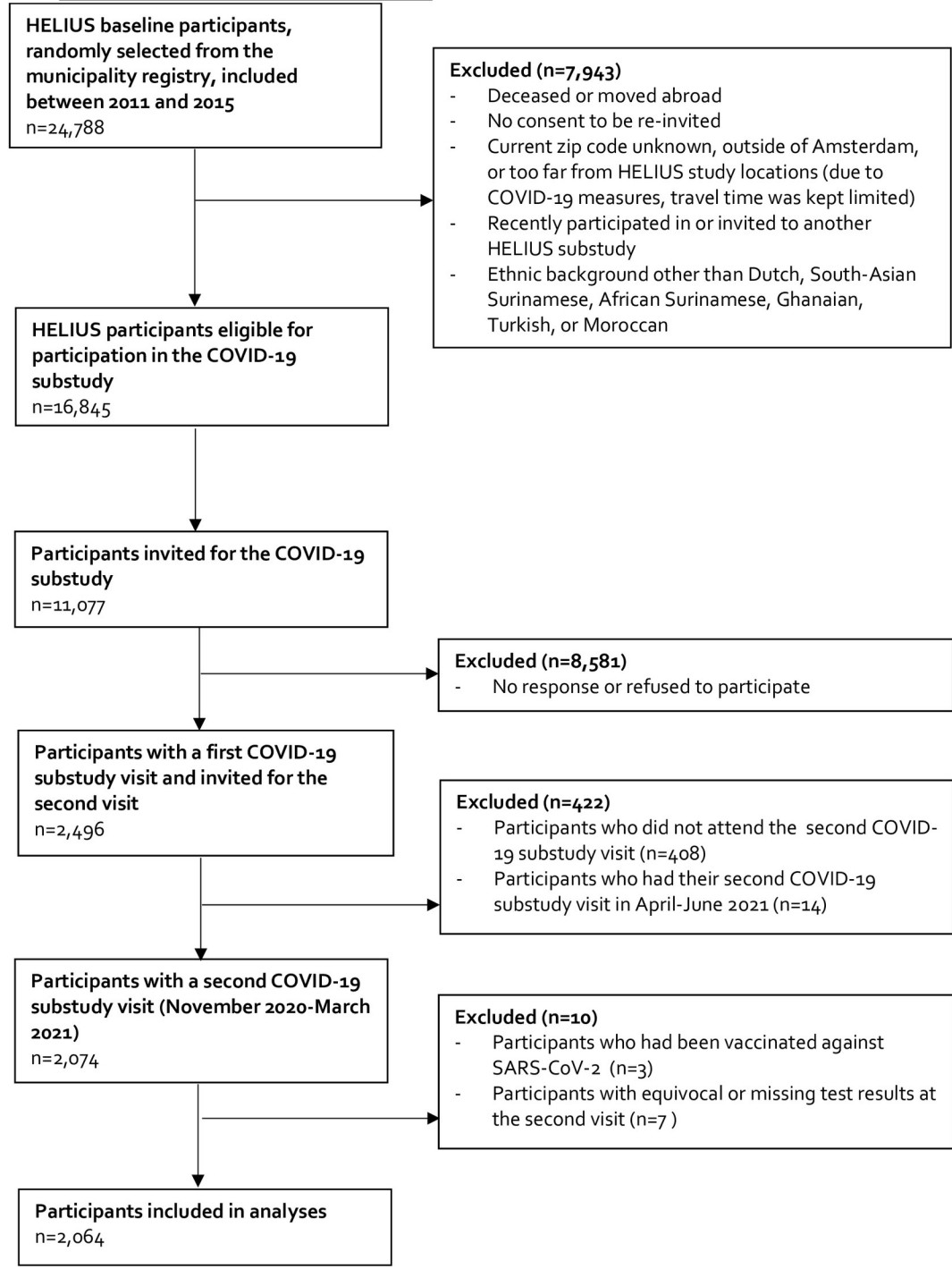

**Fig 1. Flowchart of the inclusion of HELIUS participants in the first and second visit of the longitudinal COVID-19 substudy, Amsterdam, the Netherlands.** Abbreviations: HELIUS Healthy Life in an Urban Setting; COVID-19 Coronavirus disease 2019; SARS-CoV-2 Severe Acute Respiratory Syndrome Coronavirus 2.

invited to participate in the SARS-CoV-2 serological substudy. Participants were included until the target sample of 430 per group was achieved for all ethnic groups or the recruitment period ended. In total, 2,496 participants were included at the first visit (between June 24 and October 9, 2020). At the second visit, 2,088 people returned (between November 23, 2020 and June 4, 2021). Due to the low number of participants who had their second visit after March 31, 2021, we excluded 14 participants who participated after this date, resulting in 2,074 participants. The third visit was ongoing at the time of the analyses of the current study and after vaccinations were offered to the population, and thus was not included. During all visits, serum samples for assessment of SARS-CoV-2 antibodies were collected by venipuncture and stored at −20°C. Trained interviewers asked participants questions on potential exposures to COVID-19. The first two visits took place before the roll-out of the national vaccination campaign for most age groups.

We excluded participants with equivocal (n = 2) or missing (n = 5) test results at the second visit. Finally, all participants were asked whether they were vaccinated against SARS-CoV-2. We subsequently excluded all vaccinated participants (n = 3), resulting in a study sample of 2,064 participants. The HELIUS study is carried out in accordance with the Declaration of Helsinki and has been approved by the AMC Ethical Review Board (MREC 10/100# 17.10.1729; amendment approval for SARS-CoV-2 serological substudy: MREC 2010_100#B20203). All participants provided written informed consent before participation in the study.

## SARS-CoV-2 antibodies

SARS-CoV-2-specific antibodies were determined using the WANTAI SARS-CoV-2 Ab enzyme-linked immunosorbent assay (ELISA) (Wantai Biological Pharmacy Enterprise Co., Beijing, China), according to the manufacturer's instructions. This ELISA detects IgA, IgM, and IgG against the receptor binding domain of the Sprotein of SARS-CoV-2. Signal-to-cutoff (S/CO) ratios were used to determine seropositivity. Cut-offs were the following: S/CO <0.9 (negative), S/CO > = 1.1 (positive) S/CO 0.9–1.1 (equivocal). Sensitivity and specificity of this test are high (99%) [33]. We used the test results of the second visit as the outcome measure in the analyses. All participants who had a positive test result during the first visit also had a positive test result during the second visit.

## Independent variables

Age in years at January 1, 2020, was calculated and categorized into three groups, based on an almost equal distribution of number of years per group and the general notion that the ageing process starts earlier in ethnic minority populations compared to ethnic majority populations: (1) younger (<40 years), (2) middle-aged (40–54 years), and (3) older (≥55 years).

We used job setting, occupation level, health literacy, household size, and household member with suspected infection as indicators for exposure to the SARS-CoV-2 virus [17, 27], and vitamin D intake, BMI, systolic blood pressure (BP), haemoglobin and number of comorbidities as indicators for susceptibility to infection. For all variables, we used the most recently available measurement.

## From the HELIUS baseline visit (2011–2015)

Education level, occupation level, health literacy and household size were collected at the baseline visit of the main HELIUS study. Education level was based on the self-reported highest qualification attained in the Netherlands or in the country of origin and categorized into three groups: (1) low (no, elementary only, lower vocational or lower secondary schooling), (2) intermediate (intermediate vocational, intermediate or higher secondary schooling), and (3)

high (higher vocational schooling or university). Occupation level was based on self-reported job title and job description of current or last job (in case participants were currently not working), including a question on fulfilling an executive function, and classified into three groups: (1) low (elementary, lower), (2) intermediate, and (3) high (higher or academic). Health literacy was measured using the validated set of brief screening questions (SBSQ) questionnaire [34], and categorized into adequate or inadequate. Household size was based on self-reported number of household members.

## From the HELIUS follow-up visit (2019–2020)

BMI, systolic BP, haemoglobin, hypertension, prior cardiovascular disease (CVD), diabetes, chronic kidney disease (CKD) risk, asthma/COPD were collected at the HELIUS follow-up visit. BMI was calculated as weight (kg) divided by height squared ($m^2$). Weight and height were measured in duplicate. Systolic and diastolic BP were measured in duplicate using a validated automated digital BP device (WatchBP Home; Microlife AG) in a seated position after ≥5 minutes of rest. Participants were asked to bring their prescribed medications to the research location, which were categorized using the Anatomical Therapeutic Chemical (ATC) classification system. Hypertension was defined as systolic BP ≥140 mm Hg, diastolic BP ≥90 mm Hg, use of antihypertensive medication (ATC codes C02, C03, C07, C08, C09) and/or self-reported hypertension. Fasting blood samples were drawn, and creatinin, haemoglobin, and glucose were determined. Participants were asked to bring an early morning urine sample for the analysis of albumin and creatinine levels. Participants were considered to have diabetes if they reported a diabetes diagnosis, or use of glucose-lowering medication (ATC code A10), or in case of a fasting glucose ≥7.0 mmol/l. CKD risk was categorised according to the risk of progression of kidney disease based on estimated glomerular filtration rate and albuminuria levels: (1) low, (2) moderately increased and (3) high and very high risk (merged because of low occurrence). Prior CVD was defined as self-reported prior myocardial infarction, cerebrovascular accident, angioplasty or bypass surgery (on heart or legs). Asthma/COPD was based on use of asthma or COPD medication (ATC code R03) [35]. Number of comorbidities was defined as the sum of hypertension, prior CVD, diabetes, CKD risk, and/or asthma/COPD.

## From the SARS-CoV-2 serological substudy visits (2020–2021)

Data on job setting and household member/steady partner with suspected infection were obtained from the first SARS-CoV-2 serological substudy visit. Job setting was based on self-reported occupation setting in the past year (including voluntary work and internships) and classified in three groups: (1) no job/caretaker only, (2) job with no contact within 1.5 meters, and (3) job with contact within 1.5 meters. Household member/steady partner with suspected infection (not applicable/no/yes) was based on self-report.

Self-reported vitamin D intake (yes/no) was obtained from the second SARS-CoV-2 serological substudy.

## Statistical analyses

Frequencies and percentages on sex, ethnicity, and education level were presented by age group. Within each ethnic group, we presented summary statistics (means, medians, proportions) on SARS-CoV-2 seroprevalence, education level, and all exposure and susceptibility variables by age group. Statistical differences were tested within ethnic groups using Pearson's Chi-square tests (for categorical variables), and ANOVA or Kruskal-Wallis tests (for continuous variables).

First, we compared the prevalence in SARS-CoV-2 antibodies between age groups with middle age being the reference group using a prevalence ratio (PR), which was estimated, along with its 95% confidence interval (CI), using Poisson regression with robust standard errors. This model 0 was stratified by ethnic group, and included sex and age in years to account for linear trends within age groups.

Second, to explore the extent to which age group differences in SARS-CoV-2 seroprevalence were explained by differences in factors that increase the likelihood of exposure to the virus, we added five exposure variables (i.e., job setting, occupation level, health literacy, household size, and household member with suspected infection) [17] to model 0, separately, per ethnic group. Then, we added all exposure variables that contributed substantially to age group differences to model 0, creating model 1 per ethnic group. A change of 10% or more in the estimates of age groups compared to model 0 was considered a substantial contribution of the variable, using the following formula: $(PR_{old} - PR_{new}) / (PR_{old} - 1) \times 100\%$, where PR is the prevalence ratio of the youngest or oldest age groups, compared to the middle-aged. This resulted in the inclusion of all five exposure variables in model 1 for the African Surinamese, Turkish and Moroccan groups; all except health literacy in the Dutch group; all except household member with suspected infection in the South-Asian Surinamese group; and all except job setting and household size in the Ghanaian group.

Third, we explored to what extent age group differences in SARS-CoV-2 infections were explained by differences in (1) education level, given exposure (i.e., added to model 1), and (2) vitamin D intake, BMI, systolic blood pressure, haemoglobin level, and number of comorbidities as indicators for susceptibility, given exposure (i.e., added to model 1), in separate models. Because education level may influence the models with susceptibility variables, these models were additionally adjusted for education level.

We performed three additional analyses. First, to determine whether our *a priori* selection of exposure and susceptibility variables were determinants of SARS-CoV-2 infection in our study sample, we examined in each ethnic group the associations of all explanatory variables with SARS-CoV-2 infections in separate models, adjusted for age and sex. Second, instead of the variable for number of comorbidities, we explored to what extent age group differences in SARS-CoV-2 infections were explained by differences in the individual comorbidities (hypertension, prior CVD, diabetes, CKD risk, asthma/COPD).

All statistical analyses were performed in RStudio V.4.2.1. [36] P values <0.05 were regarded as statistically significant.

## Results

The mean age of the study sample was 52.6 years (standard deviation 12.1, range 23–78 years). In most ethnic groups, the majority was 55 years or older (Table 1). Among participants of Turkish or Moroccan origin, the majority was between 40 and 54 years old. The distribution of exposure and susceptibility variables varied substantially across age groups in all ethnic groups and were in expected directions (Tables 1–3 in S1 Appendix). For instance, the younger and middle-aged groups more often reported to have a job with contact within 1.5 meter distance from others compared to the oldest group in all ethnic groups. Having two or more comorbidities was more prevalent in the oldest group compared to the other age groups in all ethnic groups.

At the second SARS-CoV-2 serological substudy visit, 495 participants had a positive antibody test result. The proportion with a positive test result was highest in the youngest age group compared to the middle-aged and older age groups in the Dutch, African Surinamese, Turkish and Moroccan groups (Fig 2, Table 4 in S1 Appendix). In the South-Asian Surinamese

**Table 1. Characteristics of participants of the first two SARS-CoV-2 serological substudy visits (2020–2021), by age groups (n = 2064).**

|  | Younger (n = 343) | Middle-aged (n = 718) | Older (n = 1003) |
|---|---|---|---|
| Sex |  |  |  |
| Male | 144 (15.8) | 289 (31.7) | 480 (52.6) |
| Female | 199 (17.3) | 429 (37.3) | 523 (45.4) |
| Ethnicity |  |  |  |
| Dutch | 76 (16.4) | 119 (25.6) | 269 (58.0) |
| South-Asian Surinamese | 61 (15.6) | 122 (31.2) | 208 (53.2) |
| African Surinamese | 30 (8.3) | 94 (26.1) | 236 (65.6) |
| Ghanaian | 20 (9.4) | 86 (40.4) | 107 (50.2) |
| Turkish | 78 (24.1) | 153 (47.2) | 93 (28.7) |
| Moroccan | 78 (25.0) | 144 (46.2) | 90 (28.8) |
| Education level |  |  |  |
| Low | 48 (14.0) | 248 (34.5) | 440 (43.9) |
| Intermediate | 124 (36.2) | 224 (31.2) | 231 (23.0) |
| High | 166 (48.4) | 232 (32.3) | 307 (30.6) |
| Missing | 5 (1.5) | 14 (1.9) | 25 (2.5) |

Younger: <40 years; middle-aged: 40–54 years; older: ≥55 years.

Data are presented as frequencies with percentages.

and Ghanaian groups, participants in the middle-aged group had the highest proportion with a positive test result.

After adjustments for age and sex (Table 2, model 0), there were no statistically significant age group differences in SARS-CoV-2 seroprevalence in the six ethnic groups (Fig 3). The direction of estimates for the association of age group with SARS-CoV-2 seroprevalence varied

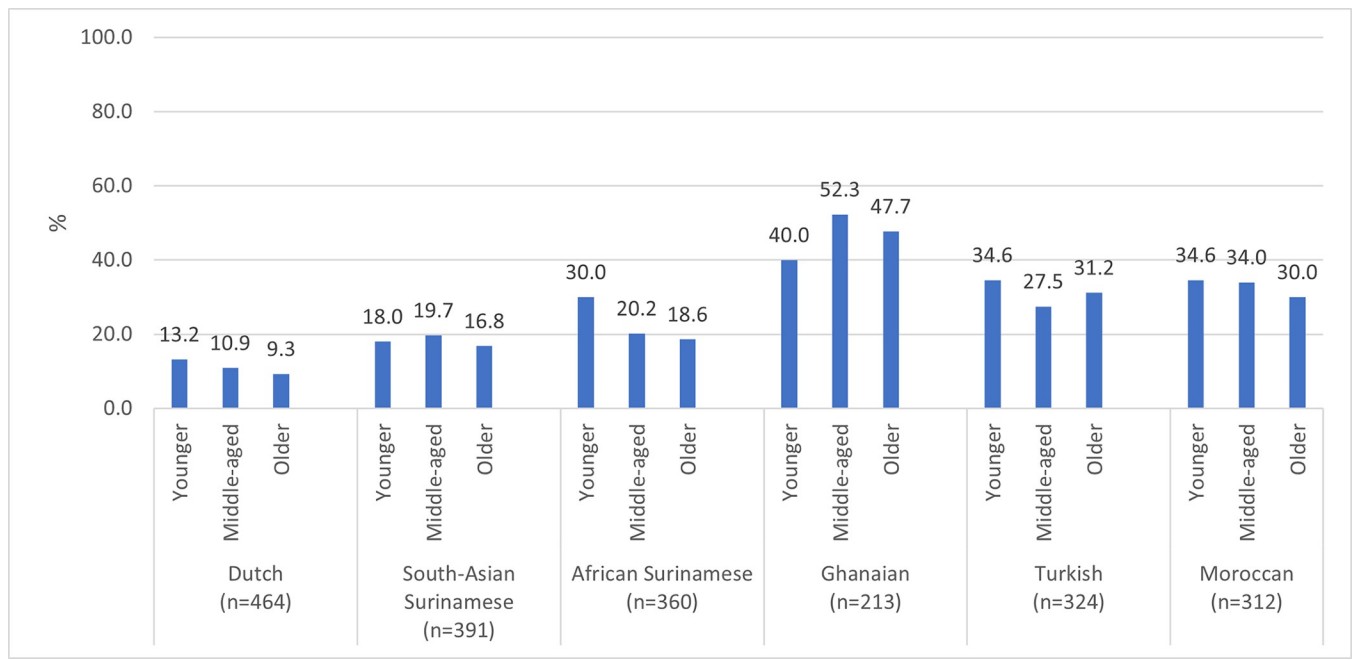

**Fig 2. Positive SARS-CoV-2 antibody test result (%), by age group and ethnicity.** Abbreviations: SARS-CoV-2 Severe Acute Respiratory Syndrome Coronavirus 2.

**Table 2. Associations between age and SARS-CoV-2 antibodies (prevalence ratios), adjusted for exposure variables, by ethnicity.**

| | Prevalence ratios (95% CI) for SARS-CoV-2 infection | | | | | |
|---|---|---|---|---|---|---|
| | **Dutch (n = 464)** | **South-Asian Surinamese (n = 391)** | **African Surinamese (n = 360)** | **Ghanaian (n = 213)** | **Turkish (n = 324)** | **Moroccan (n = 312)** |
| Model 0[a] | | | | | | |
| Younger | 2.01 (0.75–5.36) | 0.89 (0.33–2.43) | 1.25 (0.54–2.91) | 0.73 (0.36–1.51) | 1.43 (0.70–2.90) | 1.08 (0.58–2.03) |
| Middle-aged | Ref | Ref | Ref | Ref | Ref | Ref |
| Older | 0.50 (0.16–1.58) | 0.88 (0.43–1.79) | 1.11 (0.51–2.39) | 0.97 (0.60–1.59) | 1.04 (0.56–1.94) | 0.84 (0.46–1.53) |
| Model 0[a] + job setting | | | | | | |
| Younger | 2.22 (0.80–6.14) | 0.90 (0.31–2.60) | 1.33 (0.55–3.20) | 0.73 (0.36–1.49) | 1.41 (0.68–2.92) | 1.07 (0.57–2.02) |
| Middle-aged | Ref | Ref | Ref | Ref | Ref | Ref |
| Older | 0.46 (0.15–1.42) | 0.94 (0.46–1.92) | 1.05 (0.49–2.27) | 0.97 (0.59–1.59) | 1.05 (0.56–1.96) | 0.84 (0.46–1.55) |
| Model 0[a] + occupation level | | | | | | |
| Younger | 2.05 (0.74–5.68) | 0.83 (0.29–2.40) | 1.12 (0.42–2.94) | 0.56 (0.26–1.20) | 1.41 (0.63–3.18) | 1.20 (0.60–2.40) |
| Middle-aged | Ref | Ref | Ref | Ref | Ref | Ref |
| Older | 0.44 (0.13–1.48) | 0.77 (0.37–1.59) | 0.92 (0.42–2.03) | 0.96 (0.56–1.62) | 1.36 (0.65–2.84) | 0.83 (0.46–1.66) |
| Model 0[a] + health literacy | | | | | | |
| Younger | 2.03 (0.76–5.43) | 0.79 (0.30–2.13) | 1.24 (0.53–2.91) | 0.67 (0.33–1.38) | 1.34 (0.65–2.78) | 1.06 (0.56–2.01) |
| Middle-aged | Ref | Ref | Ref | Ref | Ref | Ref |
| Older | 0.49 (0.15–1.55) | 0.87 (0.43–1.74) | 1.07 (0.50–2.32) | 0.98 (0.59–1.63) | 0.94 (0.51–1.73) | 0.78 (0.41–1.47) |
| Model 0[a] + household size | | | | | | |
| Younger | **2.96 (1.07–8.18)** | 1.04 (0.37–2.90) | 1.60 (0.68–3.75) | 0.71 (0.33–1.54) | 1.63 (0.80–3.32) | 1.24 (0.63–2.45) |
| Middle-aged | Ref | Ref | Ref | Ref | Ref | Ref |
| Older | 0.52 (0.18–1.53) | 0.88 (0.44–1.75) | 0.95 (0.46–2.00) | 0.97 (0.58–1.62) | 1.04 (0.55–1.94) | 0.73 (0.40–1.31) |
| Model 0[a] + household member suspected of infection | | | | | | |
| Younger | 1.82 (0.70–4.74) | 0.88 (0.33–2.38) | 1.26 (0.53–2.97) | 0.76 (0.37–1.58) | 1.42 (0.70–2.90) | 1.03 (0.56–1.91) |
| Middle-aged | Ref | Ref | Ref | Ref | Ref | Ref |
| Older | 0.60 (0.19–1.93) | 0.89 (0.44–1.78) | 1.05 (0.50–2.21) | 0.98 (0.60–1.58) | 1.06 (0.57–1.99) | 0.89 (0.49–1.61) |
| Model 1[b] | | | | | | |
| Younger | 2.55 (0.93–6.97) | 0.79 (0.26–2.40) | 1.30 (0.49–3.47) | 0.57 (0.27–1.24) | 1.40 (0.60–3.30) | 1.27 (0.61–2.65) |
| Middle-aged | Ref | Ref | Ref | Ref | Ref | Ref |
| Older | 0.53 (0.16–1.74) | 0.82 (0.40–1.68) | 0.78 (0.36–1.69) | 0.95 (0.56–1.60) | 1.53 (0.72–3.26) | 0.89 (0.46–1.70) |

Younger: <40 years; middle-aged: 40–54 years; older: ≥55 years. Statistically significant p-values (p<0.05) are marked in bold.

[a] Adjusted for age and sex.

[b] Adjusted for age, sex and exposure variables (job setting, occupation level, health literacy, household size, household member with suspected infection). Adjustments for exposure variables differ per ethnic group

- Dutch: all, except health literacy
- South-Asian Surinamese: all, except household member with suspected infection
- Ghanaian: all, except job setting and household size
- African Surinamese, Turkish, Moroccan: all

by ethnic group. For instance, PRs pointed towards a higher seroprevalence among younger adults (<40 years) compared to middle-aged adults (40–54 years) in the Dutch (PR = 2.01, 95% CI 0.75–5.36), African Surinamese (PR = 1.25, 95% CI 0.54–2.91), Turkish (PR = 1.43, 95% CI 0.70–2.90) and Moroccan groups (PR = 1.08, 95% CI 0.58–2.03), but towards a lower seroprevalence in younger adults in the South-Asian Surinamese (PR = 0.89, 95% CI 0.33–2.43) and Ghanaian groups (PR = 0.73, 95% CI 0.36–1.51). Age group differences in seroprevalence were most pronounced in the participants of Dutch origin.

Some exposure variables, particularly health literacy, household size, household member/steady partner with suspected infection, were significantly associated with SARS-CoV-2 seroprevalence (Table 5 in S1 Appendix). Additional adjustments for these variables resulted in changes of estimates for the effect of age group (model 1), mainly towards larger differences between age groups (Fig 3 and Table 2). However, the patterns of changes were not consistent across ethnic groups. For instance, in some ethnic groups, these adjustments mainly led to a higher PR in the younger group versus middle-aged group (e.g., Dutch: PR changed from 2.01 to 2.55, and African Surinamese: from 1.25 to 1.30) but to little change in PR for the older group. The reverse was observed in other groups, such as the Turkish group (PR for the younger group compared to the middle-aged group changed from 1.43 to 1.40).

Most susceptibility variables were not significantly associated with SARS-CoV-2 seroprevalence (Table 6 in S1 Appendix). Also, overall age group patterns did not substantially change when adjusting for these variables (Fig 4A-4F) or when adjusting for the individual comorbidities (Table 7 in S1 Appendix). In addition, the effect of age group remained non-significant in most groups. However, in the Dutch group, younger adults had a significantly higher PR for SARS-CoV-2 antibodies compared to middle-aged adults after adjustments for BMI, haemoglobin level, and number of comorbidities (range of PRs: 2.79–2.83, Table 5 in S1 Appendix).

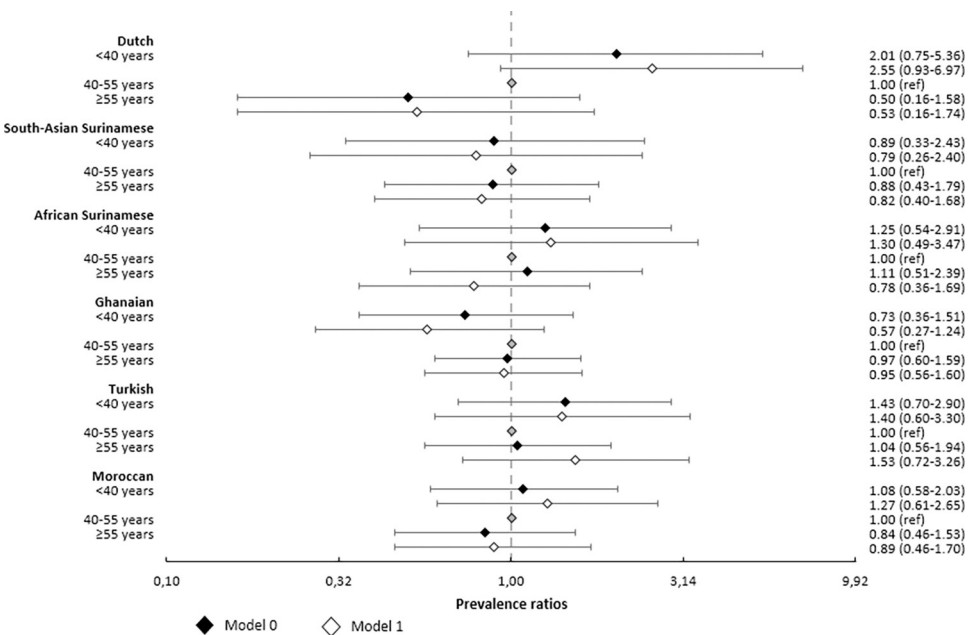

**Fig 3. Forest plot of associations between age and SARS-CoV-2 seroprevalence, age- and sex-adjusted (model 0), and exposure-adjusted (model 1), by ethnicity.**

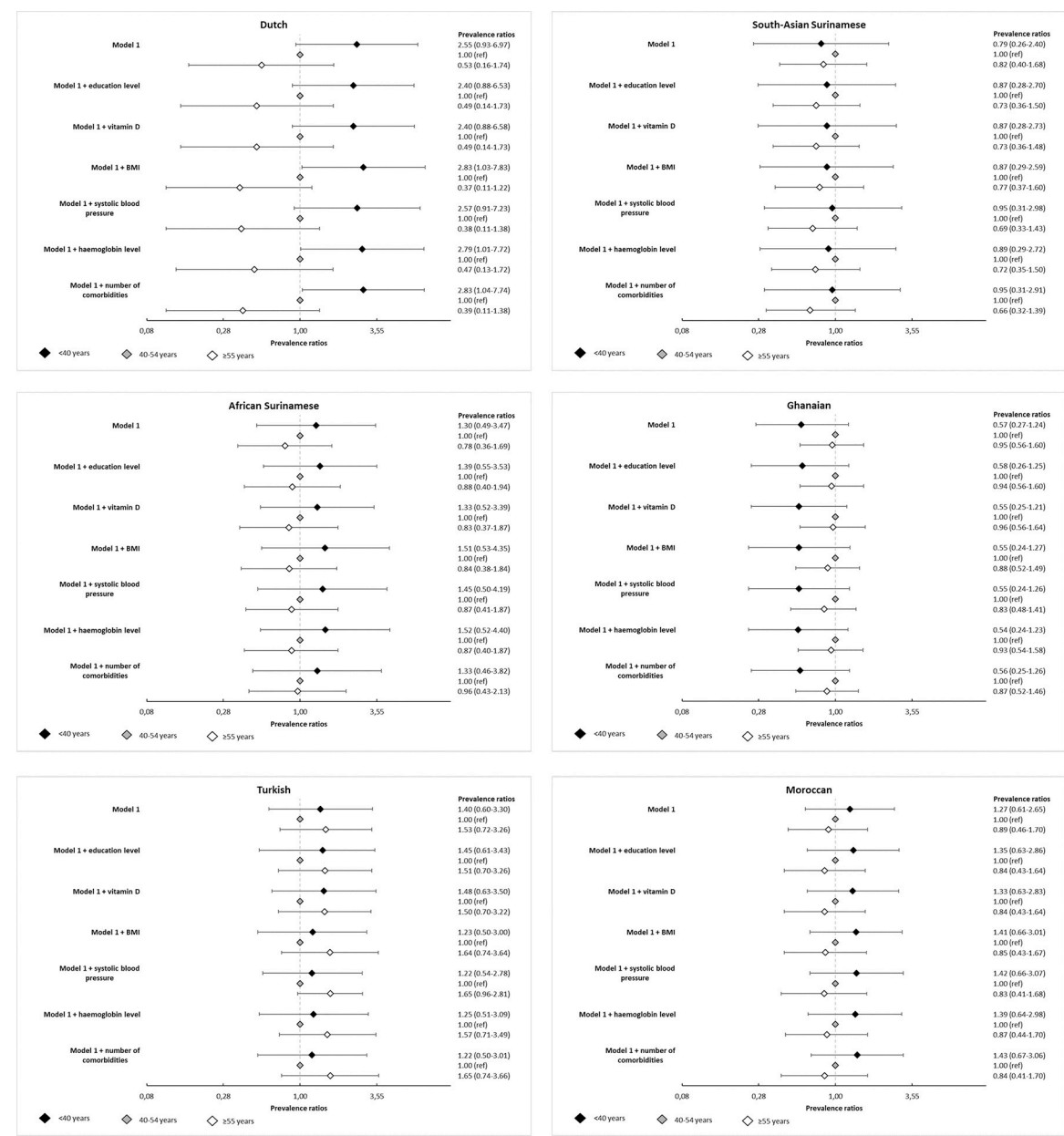

**Fig 4.** a. Forest plot of associations between age and SARS-CoV-2 seroprevalence, adjusted for age, sex and exposure (model 0), and susceptibility variables, in participants of Dutch origin. b. Forest plot of associations between age and SARS-CoV-2 seroprevalence, adjusted for age, sex and exposure (model 0), and susceptibility variables, in participants of South-Asian Surinamese origin. c. Forest plot of associations between age and SARS-CoV-2 seroprevalence, adjusted for age, sex and exposure (model 0), and susceptibility variables, in participants of African Surinamese origin. d. Forest plot of associations between age and SARS-CoV-2 seroprevalence, adjusted for age, sex and exposure (model 0), and susceptibility variables, in participants of Ghanaian origin. e. Forest plot of associations between age and SARS-CoV-2 seroprevalence, adjusted for age, sex and exposure (model 0), and susceptibility variables, in participants of Turkish origin. f. Forest plot of associations between age and SARS-CoV-2 seroprevalence, adjusted for age, sex and exposure (model 0), and susceptibility variables, in participants of Moroccan origin.

## Discussion

We found no statistically significant age group differences in SARS-CoV-2 seroprevalence among 2064 participants of the multi-ethnic HELIUS study. However, age group patterns in SARS-CoV-2 seroprevalence, i.e., irrespective of statistical significance, varied across ethnic

groups. Age group differences in SARS-CoV-2 seroprevalence were most pronounced in the Dutch majority group, with the highest occurrence of antibodies in the youngest group and the lowest in the oldest group. In the ethnic minority groups, age group differences were smaller. In all ethnic groups, these patterns did not substantially change after adjustments for exposure and susceptibility variables.

A strength of our study is the population-based sampling with a large number of participants from the major ethnic groups living in Amsterdam, the Netherlands. Furthermore, SARS-CoV-2 antibodies were measured irrespective of COVID-19-related symptoms and testing behaviour, and before the roll-out of the national vaccination campaign. In addition, exposure and susceptibility variables were measured at individual level. Nonetheless, our study has limitations.

First, the study sample was recruited from an existing cohort study and selection bias may have occurred at several points in time. For instance, at the HELIUS baseline measurement, participation and response rates varied across ethnic groups and were relatively low (50% of those contacted after invitation and 28% of those invited, respectively) [31]. Yet, although limited to age, sex and socioeconomic characteristics, non-response analyses of the HELIUS baseline measurement showed few differences between participants and non-participants across ethnic groups, suggesting that the risk of selection bias was small. In addition, loss to follow-up rates differed between ethnic groups after the baseline HELIUS study, potentially influencing representativeness of the sample for the substudy for the general Amsterdam population. However, we previously showed that the distribution of characteristics was largely similar between included and non-included HELIUS participants in the substudy [27], and that the impact of loss to follow-up was small [17]. Moreover, selective non-response may have influenced overall infection risk, but not necessarily the differences in the presence of SARS-CoV-2 antibodies between age groups, nor the determinants of these potential differences. As such, we do not think that the absence of a relationship between age group and seroprevalence rates was caused by selective non-response.

Second, the variables we used as proxy variables for exposure and susceptibility may not have measured real exposure or susceptibility, because of measurement bias or bias in self-reporting one's behaviour. Furthermore, the timing of the measurements may have been sub-optimal, even though for some variables, such as education level, it is not expected that they change frequently. For instance, the presence of SARS-CoV-2 antibodies, indicating past infection, and some of the exposure and susceptibility variables (e.g., household member with suspected infection) were measured at the same time. Thus, potential infection may have occurred before exposure. Other variables (e.g., comorbidities) were measured at the HELIUS baseline visit, up to nine years before potential infection.

Finally, our stratified analyses possibly had limited power to demonstrate statistically significant associations due to small age group sizes (particularly the younger and older groups) and the low number of positive tests (particularly the youngest group), as is suggested by the large confidence intervals. However, in an additional analysis in which we repeated the age- and sex-adjusted model (model 0) in the total study sample, i.e. without stratification by ethnicity, we also found no statistically significant associations between age groups and SARS-CoV-2 seroprevalence (data available upon reasonable request). This might indicate either that indeed there are ethnicity-specific age patterns that go in different directions, or that our non-significant findings are not only due to power issues.

We observed no significant differences in SARS-CoV-2 seroprevalence between age groups in all ethnic groups. This observation is in contrast to previous meta-analyses, in which a lower seroprevalence was reported in older adults compared to younger and middle-aged adults [2–4]. This may be related to the smaller age range of our study sample (23–78 years)

compared to most other studies. Moreover, our study was more ethnically diverse compared to other studies investigating age group differences. This is supported by our finding that age group differences are most pronounced in the Dutch majority sample, with a pattern of estimates (i.e. higher seroprevalence in the younger group and lower seroprevalence in the older group) that is in line with pooled seroprevalence in the meta-analyses based on both population-wide (household and community samples) and population-specific studies (such as healthcare workers, patient groups, and blood donors) [2–4]. In contrast, we found that SARS-CoV-2 seroprevalence is more similar across age groups in the ethnic minority groups. This explanation emphasizes the relevance of ethnic diversity in study samples, as well as the stratification of findings for multiple relevant factors (i.e., intersectionality [37]), such as age and ethnicity. Since previous studies did not stratify by both ethnicity and age group, it was not possible to verify our ethnic specific findings in other studies.

In all ethnic groups, the (lack of) age group differences were not explained by differences in exposure and susceptibility. This is particularly remarkable for the exposure variables as they are considered the most important mechanism for infection [5]. Moreover, we did observe substantial age group differences in exposure across ethnic groups. As mentioned, it might be that our measurement of the exposure variables was suboptimal, or that we have missed some important exposure factors (such as wearing face masks, keeping distance). In any case, our findings support the need for more research to better understand the variation in age group patterns in presence of SARS-CoV-2 antibodies across ethnic groups and the explanatory factors associated with this variation.

In mid-March 2020, at the beginning of the COVID-19 epidemic in the Netherlands, the Dutch government implemented several extensive measures to contain the further spread of the SARS-CoV-2 virus [38]. In addition, older adults and vulnerable people (e.g., chronically ill) were specifically advised to prevent exposure to the virus, because of their higher risk of severe COVID-19 and death, by e.g. keeping distance from other people as much as possible. Our finding of a lower seroprevalence among older adults of the Dutch majority sample may reflect that these measures and advices were able to prevent the spread of SARS-CoV-2 virus particularly among the older population of Dutch origin and less so among that of other ethnic origins. The fact that the seroprevalence is more similar across age groups in the ethnic minority groups may be related to an overall lower uptake of the preventive measures among the ethnic minority groups [39] and to higher risk of exposure due to, for instance, job setting and dependency on public transport [24, 40], irrespective of age. However, a previous analysis among a selection of HELIUS participants suggests that adherence to some of these measures was similar to Dutch origin participants [41]. This finding indicates that the varying age group patterns between ethnic groups are likely not explained by differences in adherence to measures in our study sample. It may, however, be related to more structural inequities between groups leading to higher risk of exposure among ethnic minority groups [21]. It was indeed found that, next to migration background, infection rates in Amsterdam were also independently associated with lower SES city districts [42].

It is striking that older adults among the ethnic minority groups were as likely as younger and middle-aged adults to have had a SARS-CoV-2 infection in the first waves of the pandemic. This group may be particularly vulnerable due to their higher risk of severe disease and mortality after a SARS-CoV-2 infection because of both older age [6–8] and their ethnic minority status [43, 44]. Thus, for the future out roll of targeted prevention activities in outbreaks of COVID-19 and other infections with a major impact on the health of older individuals, it is important to realise that older adults in ethnic minority groups may be at an increased risk. This will be especially relevant if it would turn out that, due to lower vaccine uptake [45],

COVID-19 vaccination has not been able to eliminate, or at least diminish, this higher susceptibility.

In conclusion, we found no evidence for age group differences in SARS-CoV-2 infections in a multi-ethnic sample aged 23–78 years, irrespective of factors related to exposure to the virus or susceptibility to infection. However, patterns of age group differences appeared to vary across ethnic groups, with a more pronounced pattern observed in the Dutch group (i.e., highest seroprevalence in the youngest group and the lowest in the oldest group), compared to the ethnic minority groups. Our findings based on measurements of antibodies against SARS-CoV-2 before the roll-out of the national vaccination campaign may thus imply that, given the fact that infections are more common among minority groups [17], older adults of the five ethnic minority groups in our study (South-Asian Surinamese, African Surinamese, Ghanaian, Turkish, Moroccan) are an important target group for additional prevention strategies for future infectious disease outbreaks.

## Supporting information

**S1 Appendix. Supplemental material to 'Differences in SARS-CoV-2 antibody prevalence at the end of the pre-vaccination period between age groups in a multi-ethnic population: A cross-sectional analysis of the population-based HELIUS study'.**
(DOCX)

## Acknowledgments

We are most grateful to the participants of the HELIUS study and the management team, research nurses, interviewers, research assistants and other staff who have taken part in gathering the data of this study.

## Author Contributions

**Conceptualization:** Renee Bolijn, Annemieke M. W. Spijkerman, Henrike Galenkamp, Maria Prins, Karien Stronks.

**Data curation:** Henrike Galenkamp, Liza Coyer.

**Formal analysis:** Renee Bolijn.

**Methodology:** Renee Bolijn.

**Supervision:** Henrike Galenkamp, Karien Stronks.

**Validation:** Henrike Galenkamp, Liza Coyer.

**Visualization:** Renee Bolijn.

**Writing – original draft:** Renee Bolijn.

**Writing – review & editing:** Renee Bolijn, Annemieke M. W. Spijkerman, Henrike Galenkamp, Anneke Blokstra, Liza Coyer, Anders Boyd, Maria Prins, Karien Stronks.

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
