## [Decision Letter · Decision Letter 0]

3 Jul 2024

PONE-D-24-05754Differences in SARS-CoV-2 antibody prevalence at the end of the pre-vaccination period between age groups : a cross-sectional analysis of the multi-ethnic population-based HELIUS studyPLOS ONE

Dear Dr. Galenkamp,

Thank you for submitting your manuscript to PLOS ONE. After careful consideration, we feel that it has merit but does not fully meet PLOS ONE’s publication criteria as it currently stands. Therefore, we invite you to submit a revised version of the manuscript that addresses the points raised during the review process (see below).

We look forward to receiving your revised manuscript.

Kind regards,

Andrey I Egorov

Academic Editor

PLOS ONE

“National Institute for Public Health and the Environment (RIVM) as part of the COVID-19 research programme commissioned by the Ministry of Health, Welfare and Sports.”

3. For studies involving third-party data, we encourage authors to share any data specific to their analyses that they can legally distribute. PLOS recognizes, however, that authors may be using third-party data they do not have the rights to share. When third-party data cannot be publicly shared, authors must provide all information necessary for interested researchers to apply to gain access to the data. (https://journals.plos.org/plosone/s/data-availability#loc-acceptable-data-access-restrictions)

4) All necessary contact information others would need to apply to gain access to the data.

Reviewers' comments:

Reviewer's Responses to Questions

**Comments to the Author**

1. Is the manuscript technically sound, and do the data support the conclusions?

Reviewer #1: Partly

Reviewer #2: Partly

2. Has the statistical analysis been performed appropriately and rigorously? 

Reviewer #1: N/A

Reviewer #2: Yes

3. Have the authors made all data underlying the findings in their manuscript fully available?

Reviewer #1: Yes

Reviewer #2: Yes

4. Is the manuscript presented in an intelligible fashion and written in standard English?

Reviewer #1: Yes

Reviewer #2: Yes

5. Review Comments to the Author

Reviewer #1: a. The risk of SARS-CoV-2 infection will always be higher in most ethnic minority groups across the globe because of their economic status and their exposure risk at work place. Therefore, I think that the exposure risk and disease risk should be assessed and compared only with variables such as economic status, outdoor activities, work environment, availability of PPEs etc. not with ethnicity.

b. The age wise comparison and interpretation is also not very sound in the context of people with <40 yrs are in productive age and are part of workforce in nay nation, therefore as this group is more active and defiant of various controls imposed, they end up with more exposure than older people with less activity and more comfortable with rules and controls. Authors need to revisit their findings in light of this.

c. All the interpretations of the study must be done after adjustment for confounders and explanatory factors of SARS-Cov-2 exposure and disease.

d. All the advanced serological assays which have been designed, validated, and tested for SARS-Cov-2 are based on accurate quantitation of the humoral response post natural infection.

e. Significant increase in antibody titer and epitope coverage occurs with increasing age which is an established fact in both mild and severe disease.

Please refer the following articles:

1) Martin CA, Pan D, Melbourne C, Teece L, Aujayeb A, Baggaley RF, Bryant L, Carr S, Gregary B, Gupta A, Guyatt AL, John C, McManus IC, Nazareth J, Nellums LB, Reza R, Simpson S, Tobin MD, Woolf K, Zingwe S, Khunti K, Abrams KR, Gray LJ, Pareek M; UK-REACH Study Collaborative Group. Risk factors associated with SARS-CoV-2 infection in a multiethnic cohort of United Kingdom healthcare workers (UK-REACH): A cross-sectional analysis. PLoS Med. 2022 May 26;19(5):e1004015. doi: 10.1371/journal.pmed.1004015. PMID: 35617423; PMCID: PMC9187071.

2) Patel M, Nair M, Pirozzoli E, Cienfuegos MC, Aitken E. Prevalence and socio-demographic factors of SARS-CoV-2 antibody in multi-ethnic healthcare workers. Clin Med (Lond). 2021 Jan;21(1):e5-e8. doi: 10.7861/clinmed.2020-0619. PMID: 33479076; PMCID: PMC7850215.

3) Martin CA, Woolf K, Bryant L, Goss C, Gogoi M, Lagrata S, Papineni P, Qureshi I, Wobi F, Nellums L, Khunti K, Pareek M; UK-REACH Study Collaborative Group. Coverage, completion and outcomes of COVID-19 risk assessments in a multi-ethnic nationwide cohort of UK healthcare workers: a cross-sectional analysis from the UK-REACH Study. Occup Environ Med. 2023 Jul;80(7):399-406. doi: 10.1136/oemed-2022-108700. Epub 2023 May 23. PMID: 37221040; PMCID: PMC10314065.

4) Siddiq S, Ahmed S, Akram I. Clinical outcomes following COVID-19 infection in ethnic minority groups in the UK: a systematic review and meta-analysis. Public Health. 2023 Sep;222:205-214. doi: 10.1016/j.puhe.2022.05.019. Epub 2022 Jun 9. PMID: 35970621; PMCID: PMC9181265.

5) Irizar P, Pan D, Kapadia D, Bécares L, Sze S, Taylor H, Amele S, Kibuchi E, Divall P, Gray LJ, Nellums LB, Katikireddi SV, Pareek M. Ethnic inequalities in COVID-19 infection, hospitalisation, intensive care admission, and death: a global systematic review and meta-analysis of over 200 million study participants. EClinicalMedicine. 2023 Mar;57:101877. doi: 10.1016/j.eclinm.2023.101877. Epub 2023 Mar 6. PMID: 36969795; PMCID: PMC9986034.

6) Agyemang C, Richters A, Jolani S, Hendriks S, Zalpuri S, Yu E, Pijls B, Prins M, Stronks K, Zeegers MP. Ethnic minority status as social determinant for COVID-19 infection, hospitalisation, severity, ICU admission and deaths in the early phase of the pandemic: a meta-analysis. BMJ Glob Health. 2021 Nov;6(11):e007433. doi: 10.1136/bmjgh-2021-007433. PMID: 34740916; PMCID: PMC8573300.

Reviewer #2: In the work "Differences in SARS-CoV-2 antibody prevalence at the end of the pre-vaccination period between age groups : a cross-sectional analysis of the multi-ethnic population- based HELIUS study" authors investigated SARS-CoV-2 seroprevalence and did not find statistically significant differences across age groups. Particular focus of their study is ethnic component and investigation of the age differences for seroprevalence.

I found the study interesting and important for the field. The core minus is that the data is out of hot topic and belong to the first part of the COVID-19 pandemic.

However, I have few major comments which I believe should be satisfied in order to make the manuscript accepted for publication.

1. It would be good if authors remake the figures to better point differences not only for age, but also across ethnic groups. Possibly putting them on one plot or choosing any other way that data for age, ethnicity can be on the same plot to allow quantitative comparison between groups

2. I recommend to complement the manuscript with more quantitative information in order to make the text and data maximally useful for the epidemiology community

3. Better elaboration of the current data in terms of seroconversion, maybe comparison with other infectious diseases would make the manuscript much better and useful for scientific community. Because the topic authors explore in the manuscript is old, it is necessary to present information in the context of the current epidemic status and possible future dangers

4. It would be great to see comparison with other studies, especially in Asia. Moreover, would be great to compare results for the ethnic groups in the Netherlands and their home countries.

5. I would recommend statistical information in the tables transfer to the visual from, possibly keeping tables for the supplementary part.

6. Is it possible to provide also geospatial characteristics of the seroconversion? If the data are available, certain local differences even within the same ethnic group can be interesting, especially in the context of pandemic management considering local peculiarities. The Dutch group is the largest and spread over the whole country, but what about other ethnic groups? Do we see differences between small towns and large cities?

7. In general, I would recommend more focus on the visual representation of the data in order to present the conclusions in the best possible way for the reader.

6. PLOS authors have the option to publish the peer review history of their article (what does this mean?). If published, this will include your full peer review and any attached files.

Reviewer #1: **Yes: **Dr. Suresh Yadav

Reviewer #2: No

---

## [Author Response · Author response to Decision Letter 0]

29 Aug 2024

Dear Editor,

We thank all reviewers for their valuable comments on our manuscript. In the response to reviewers document, we respond to each of the issues raised by the editor and reviewers. Each comment is numbered and a reference is given to the corresponding part of the paper that has been revised. Please note that the page and line numbers correspond to those in the tracked changes version.

Kind regards,

Henrike Galenkamp

---

## [Decision Letter · Decision Letter 1]

16 Sep 2024

Differences in SARS-CoV-2 antibody prevalence at the end of the pre-vaccination period between age groups : a cross-sectional analysis of the multi-ethnic population-based HELIUS study

PONE-D-24-05754R1

Dear Dr. Galenkamp,

We’re pleased to inform you that your manuscript has been judged scientifically suitable for publication and will be formally accepted for publication once it meets all outstanding technical requirements.

Kind regards,

Andrey I Egorov

Academic Editor

PLOS ONE

Reviewers' comments:

Reviewer's Responses to Questions

**Comments to the Author**

1. If the authors have adequately addressed your comments raised in a previous round of review and you feel that this manuscript is now acceptable for publication, you may indicate that here to bypass the “Comments to the Author” section, enter your conflict of interest statement in the “Confidential to Editor” section, and submit your "Accept" recommendation.

Reviewer #1: All comments have been addressed

Reviewer #2: All comments have been addressed

2. Is the manuscript technically sound, and do the data support the conclusions?

Reviewer #1: Yes

Reviewer #2: Yes

3. Has the statistical analysis been performed appropriately and rigorously? 

Reviewer #1: Yes

Reviewer #2: Yes

4. Have the authors made all data underlying the findings in their manuscript fully available?

Reviewer #1: Yes

Reviewer #2: Yes

5. Is the manuscript presented in an intelligible fashion and written in standard English?

Reviewer #1: Yes

Reviewer #2: Yes

6. Review Comments to the Author

Reviewer #1: The exposure variables such as health literacy, household size, and household member/steady partner with suspected infection, etc have been quantitatively, presented and association was found with SARS-CoV-2 seroprevalence, the authors have done great work by using the model. I am satisfied with the efforts and replies to the queries raised by the reviewers.

Reviewer #2: Thank to the authors for addressing the raised issues. Modifications they made improved the manuscript. I hope the results will be useful for the scientific community.

7. PLOS authors have the option to publish the peer review history of their article (what does this mean?). If published, this will include your full peer review and any attached files.

Reviewer #1: **Yes: **Suresh Yadav

Reviewer #2: No

---

## [Editor Report · Acceptance letter]

27 Sep 2024

PONE-D-24-05754R1 

PLOS ONE

Dear Dr. Galenkamp, 

I'm pleased to inform you that your manuscript has been deemed suitable for publication in PLOS ONE. Congratulations! Your manuscript is now being handed over to our production team.

Kind regards, 

on behalf of

Dr. Andrey I Egorov 

Academic Editor

PLOS ONE